# LARGE LANGUAGE MODELS ARE INNATE CRYSTAL STRUCTURE GENERATORS

**Jingru Gan,**[*] **Yanqiao Zhu, Daniel Schwalbe-Koda & Wei Wang**
University of California, Los Angeles
Los Angeles, CA, USA

**Peichen Zhong & Kristin A. Persson**
University of California, Berkeley
Berkeley, CA, USA

**Yuanqi Du & Carla P. Gomes**
Cornell University
Ithaca, NY 14853, USA

**Haorui Wang**
Georgia Institute of Technology
Atlanta, GA 30332, USA

**Chenru Duan**
Deep Principle Inc.
Dover, DE 19901, USA

## ABSTRACT

Crystal structure generation is fundamental to materials discovery, enabling the prediction of novel materials with desired properties. While existing approaches leverage Large Language Models (LLMs) through extensive fine-tuning on materials databases, we show that pre-trained LLMs can inherently generate stable crystal structures without additional training. Our novel framework MATLLMSEARCH integrates pre-trained LLMs with evolutionary search algorithms, achieving a 78.38% metastable rate validated by machine learning interatomic potentials and 31.7% DFT-verified stability via quantum mechanical calculations, outperforming specialized models such as CrystalTextLLM. Beyond crystal structure generation, we further demonstrate that our framework can be readily adapted to diverse materials design tasks, including crystal structure prediction and multi-objective optimization of properties such as deformation energy and bulk modulus, all without fine-tuning. These results establish pre-trained LLMs as versatile and effective tools for materials discovery, opening up new venues for crystal structure generation with reduced computational overhead and broader accessibility.

## 1 INTRODUCTION

Discovering materials with desired properties remains a fundamental challenge in materials science. The critical step is predicting thermodynamically stable crystal structures, which determine the physical and chemical characteristics of a material [5]. While experimental synthesis and characterization remain the gold standard, computational approaches have emerged as indispensable tools for accelerating materials discovery [19, 20]. The field has evolved from evolutionary algorithms to deep learning approaches. Early evolutionary algorithms provide effective strategies for exploring the vast chemical space of possible structures [2], enabling automated property-guided materials optimization. Recent advances in deep learning have introduced various generative models for structure prediction, ranging from variational autoencoders that learn compact crystalline representations to diffusion and flow models for direct atomic configuration sampling [21, 24, 29, 53, 58]. These models employ graph neural networks to capture complex many-body interactions and crystallographic symmetries.

More recently, Large Language Models (LLMs) have emerged as promising tools for crystal structure generation [1, 4, 22]. The seminal work by Flam-Shepherd & Aspuru-Guzik [21] demonstrates that auto-regressive models with character-level tokenization can generate chemically valid crystal structures. Subsequent work [24] shows that fine-tuning pre-trained language models like Llama [23] on materials datasets can produce physically stable crystal structures. Given the vast scientific corpora that LLMs are pre-trained on, we hypothesize that these models already possess rich chemical knowledge that could enable direct crystal structure generation, eliminating the the computational overhead of specialized fine-tuning. To verify this, we pose a challenging question: *Can pre-trained LLMs be directly used to generate stable crystal structures without additional fine-tuning?*

---

[*]Corresponding author: jrgan@cs.ucla.edu

While promising, leveraging pre-trained LLMs for crystal structure generation faces several challenges: guiding the LLMs to output valid crystal structure representations, preserving crystallographic constraints in proposed structures, and ensuring thermodynamically stability of final configurations. To address them, we introduce MATLLMSEARCH, a novel framework that synergistically integrates the chemical space exploration capabilities of evolutionary algorithms with the rich chemical knowledge embedded in pre-trained LLMs. As illustrated in Figure 1, our framework implements an iterative pipeline with three key stages: (1) **Selection** identifies promising candidate structures to guide subsequent generations, (2) **Reproduction** guides LLMs in breeding new candidates from parent structures via implicit crossover and mutations, and (3) **Evaluation** enforces crystallographic constraints and assesses thermodynamic stability through a comprehensive validation pipeline.

Through comprehensive experiments, we show that our framework successfully generates diverse, thermodynamically stable crystal structures while maintaining crystallographic validity. Guided by MATLLMSEARCH, the LLM achieves a 76.81% metastable structure generation rate, with 31.70% of structures verified as stable through DFT calculations, surpassing the state-of-the-art fine-tuned model CrystalTextLLM [24]. Notably, this performance is achieved with minimal computational overhead, requiring only LLM inference and stability evaluation rather than extensive model training. Also, we only use thousands of reference structures, while CrystalTextLLM requires fine-tuning on the full Materials Project database of 45,231 stable structures [28].

Beyond crystal structure generation, our framework demonstrates remarkable flexibility across various materials discovery tasks. Through simple modifications in prompting and reference structure selection criteria, our method extends to crystal structure prediction, which is validated by the discovery of several metastable $Na_3AlCl_6$ polymorphs with significantly higher stability than existing structures in the Materials Project database. Furthermore, the framework enables multi-objective optimization of properties such as bulk modulus, suggesting its versatility across the spectrum of materials discovery challenges.

## 2 BACKGROUND: COMPUTATIONAL MATERIALS DISCOVERY WITH MACHINE LEARNING

### 2.1 PROBLEM DEFINITION

**Crystal Structure Generation (CSG).** The objective of CSG is to learn a probability distribution $p(c, l, s)$ over crystalline materials, where $c \in \mathbb{R}^{N \times K}$ represents the chemical composition matrix for $N$ atoms of $K$ distinct chemical species, $l \in \mathbb{R}^6$ denotes the lattice parameters (lengths and angles), and $s \in \mathbb{R}^{N \times 3}$ defines the spatial coordinates of atoms within a periodic unit cell. Samples drawn from this distribution should ideally satisfy fundamental thermodynamic stability criteria (defined in Section 2.2).

**Crystal Structure Prediction (CSP).** CSP addresses a more constrained problem of determining stable crystal structures for a specified chemical composition. Formally, it learns a conditional probability distribution $p(s, l \mid c)$ to identify thermodynamically favorable atomic arrangements and lattice parameters given a fixed composition $c$. This formulation addresses the practical scenario of discovering stable polymorphs for a specified chemical formula.

**Crystal Structure Design (CSD).** CSD extends beyond structure prediction by incorporating property optimization and conditional generation. An example objective is finding the optimal crystal structure that maximizes a target property $h(c, l, s)$: $m^* = \mathrm{argmax}_{c,l,s \sim p(c,l,s)} h(c, l, s)$, where $h : \mathbb{R}^{N \times K} \times \mathbb{R}^6 \times \mathbb{R}^{N \times 3} \to \mathbb{R}$ represents an oracle function evaluating the desired materials property. It can also be formulated as sampling from a tilted distribution $p(c, l, s) \exp(h(c, l, s))$ [44]. Additional constraints can be integrated into the design process, allowing for flexible tasks such as compositional substitution (learning $p(c \mid l, s)$) and composition/structure completion (inpainting generation, learning $p(c^{\mathrm{unknown}}, s^{\mathrm{unknown}} \mid c^{\mathrm{known}}, l, s^{\mathrm{known}})$) [15].

### 2.2 (META)STABILITY OF MATERIALS

Among computational approaches for evaluating crystal structure stability, Density Functional Theory (DFT) calculations stand as the most reliable method for predicting formation energies in solid-state

materials, showing close alignment with experimental measurements [27, 49]. The thermodynamic stability of a structure is quantified through its decomposition energy ($E_d$) with respect to the convex hull of known stable phases: $E_d = E_s - \sum_i x_i E_i$, where $E_s$ represents the total energy per atom, $x_i$ denotes the molar fraction of the $i$-th competing phase, and $E_i$ corresponds to its ground-state energy per atom. While the convex hull serves as a fixed reference, the evaluated structure $s$ need not be part of this hull. A negative decomposition energy ($E_d < 0$) indicates a thermodynamically stable state below the convex hull, while $E_d > 0$ suggests a metastable phase with a driving force for decomposition into more stable compounds. Our main objective is to identify stable crystal structures where $E_d \leq 0$.

Given the computational intensity of DFT calculations, universal Machine Learning Interatomic Potentials (MLIPs), trained on millions of DFT calculations, have emerged as efficient and reliable proxies for structure stability assessment. Notable among these is CHGNet [16], a Graph Neural Network (GNN)-based MLIP that uniquely incorporates magnetic moments to capture both atomic and electronic interactions. M3GNet [10] offers an alternative approach, implementing three-body interactions in its graph architecture for accurate structural predictions across diverse chemical spaces. Recent advances in universal MLIPs include MACE [7], DPA-1 [59], and JMP [47], which demonstrate high accuracy in predicting crystal thermodynamic stability, particularly when trained on industrial-scale datasets comprising millions of compounds and non-equilibrium atomic configurations [6, 37, 55]. In this work, we employ the pre-trained CHGNet as our universal MLIP due to its closer alignment with DFT results, using a fixed phase diagram derived from the Materials Project 2023 DFT calculations [27, 50].

## 3  MATLLMSEARCH

We propose MATLLMSEARCH, an evolutionary workflow that leverages pre-trained LLMs to search for stable and optimized crystal structures with. In this section, we introduce three key stages of the workflow as illustrated in Figure 1: (1) **Selection**, which identifies promising candidate structures from existing pools based on stability and property metrics; (2) **Reproduction**, where the LLM generates new candidates through implicit crossover and mutations of parent structures; and (3) **Evaluation**, which assesses proposed structures for validity, stability, and target properties. The overall workflow, outlined in Algorithm 1, iteratively evolves a population of crystal structures while maintaining physical constraints and optimizing desired properties.

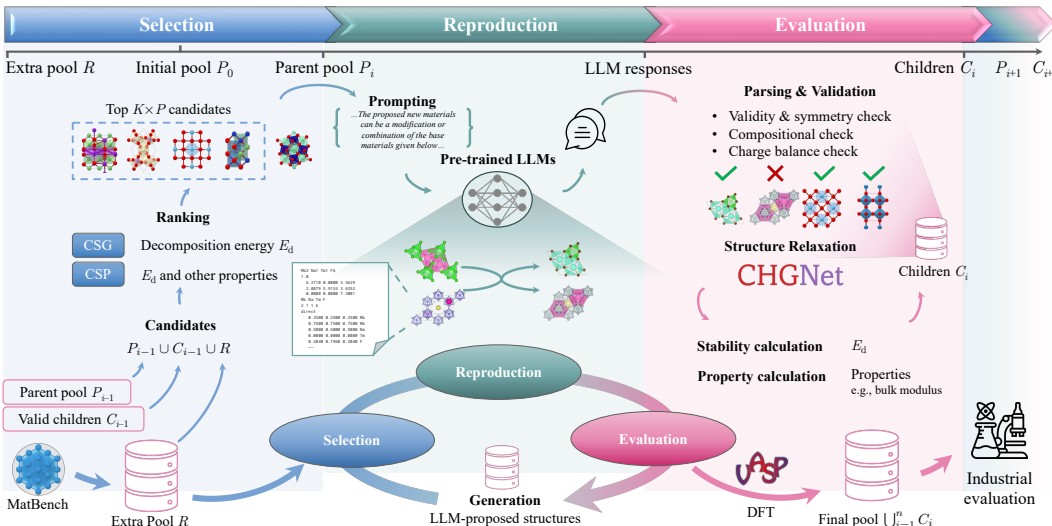

Figure 1: The workflow of MATLLMSEARCH for crystal structure generation. Starting from an initial population of known structures, our framework iteratively evolves new crystal structures through LLM-guided reproduction, evaluation, and selection.

## 3.1 INITIALIZATION

Our evolutionary search begins by constructing a diverse and valid starting population. We sample $(K \times P)$ structures from a set of known stable structures $\mathcal{D}$ to form our initial parent pool $\mathcal{P}_0$, where $K$ is the population size and $P$ is the number of parent structures per group. These structures are organized into $K$ groups of $P$ parents each to serve as reference examples in LLM prompts, with the LLM being queried $K$ times to generate new candidate structures. This grouping strategy enables the LLM to analyze multiple reference structures simultaneously when proposing new candidates. Optionally, we can retrieve an extra pool of structures $\mathcal{R}$ from $\mathcal{D}$ to expand the candidate space during the selection stage. $\mathcal{R}$ can be customized to suit various design objectives, with more details and ablation studies provided in Section 4.3. The initialization parameters and detailed sampling strategy are described in Section 4.1.

## 3.2 REPRODUCTION

Genetic algorithms traditionally mimic biological evolution through explicit crossover and mutation operations [25, 30]. In crystal structure prediction, crossover typically involves combining structural fragments from parent structures (e.g., swapping atomic positions or structural motifs), while mutation introduces random variations through predefined operations like atomic displacement, lattice transformation, or element substitution [14, 31]. While effective, these rigid operators can limit the exploration of the complex crystal structure space. In MATLLMSEARCH, we explore the flexibility of LLMs for structure reproduction. Through prompt-based guidance, we ask LLMs to perform implicit crossover and mutation by analyzing and combining structural information from parent materials. Specifically, LLMs are instructed to "modify or combine the base materials", while maintaining chemical validity and enhancing target properties. This approach allows LLMs to freely and simultaneously introduce variations across multiple structural aspects, including atomic positions, lattice parameters, and element substitutions, or even generate completely new structures functionally relevant to parent structures.

## 3.3 EVALUATION

In genetic algorithms, evaluation serves as a crucial bridge between reproduction and selection by assessing the fitness of offspring structures. Following LLM-guided reproduction, we employ a two-stage evaluation pipeline to validate and evaluate the generated structures and ensure they represent physically meaningful candidates for the next generation. Specifically, the evaluation process integrates rule-based filters for fundamental physical constraints and quantitative stability metrics, with optional additional property calculations to assess candidate performance.

**Rule-based structure validation.** We first apply a series of basic criteria to validate structural integrity. Each parsed structure is extracted from LLM responses into standardized crystallographic formats and must satisfy fundamental physical requirements, most importantly three-dimensional periodicity with proper boundary conditions. Then, physical connectivity is ensured by requiring valid bonding for each atom, defined as interatomic distances between 0.6 to 1.3 times the sum of constituent atomic radii. Chemical validity is verified through charge balance analysis based on formal valence states of the constituent elements. To maintain structural diversity in the population, duplicate structures generated within the same iteration are eliminated.

**Stability and property evaluation.** Children structures that satisfy the rule-based validation will undergo evaluation of stability and other specific target properties based on the design objectives. Since LLM-proposed structures may not be at their local energy minimum, each structure is first relaxed using CHGNet. We monitor the energy difference $\Delta E$ between relaxed and initial states, where a larger $|\Delta E|$ indicates the initial structure required more significant relaxation to reach stability. Notably, we show that LLM-proposed structures typically require minimal relaxation, with 61.1% of structures exhibiting small energy changes ($|\Delta E| < 0.5$ eV/atom) during this process (detailed in Appendix H). The choice of evaluation metrics depends on the optimization objectives. For stability-focused optimization, we quantify thermodynamic stability through the decomposition energy $E_d$ using CHGNet, calculated as the distance to the convex hull from the Materials Project database (version `2023-02-07-ppd-mp`). For mechanical property-oriented objectives, other properties such as bulk modulus can be computed in this stage. These quantitative scores then guide the selection process for subsequent generations, allowing flexible adaptation to different design goals.

| Model | $f$-ele in Parents[†] | Validity | | Metastability | | | Stability[‡] | |
|---|---|---|---|---|---|---|---|---|
| | | Structural | Composition | M3GNet | CHGNet | | DFT | |
| | | | | $E_d < 0.1$ | $E_d < 0.1$ | $E_d < 0.03$ | w/ $f$-ele | w/o $f$-ele[§] |
| CDVAE* | — | 100.0% | 86.7% | 28.8% | — | — | 5.4% | — |
| CrystalTextLLM-7B* | — | 96.4% | 93.3% | 35.0% | — | — | 8.7% | — |
| CrystalTextLLM-13B* | — | 95.5% | 92.4% | 38.0% | — | — | 14.4% | — |
| CrystalTextLLM-70B* | — | 99.6% | 95.4% | 49.8% | — | — | 10.6% | — |
| MATLLMSEARCH | ✓ | 100.0% | 79.4% | 81.1% | 76.8% | 56.5% | 31.7% | 14.0% |
| (Llama 3.1-70B) | ✗ | 100.0% | 89.0% | 81.9% | 78.4% | 54.8% | 27.0% | 24.6% |

Table 1: Performance comparison of crystal structure generation. Metastability is first assessed using surrogate models, where we report both M3GNet and CHGNet results for fair comparison with baselines CDVAE and CrystalTextLLM (which use M3GNet). *Results taken from the original papers. [†]Indicates whether $f$-electron elements are excluded in parent structures (not applicable to CDVAE and CrystalTextLLM as they are trained on data including $f$-electron elements). [‡]The stable fraction represents the percentage of DFT-verified stable structures ($E_d < 0.0$ eV/atom) over structures predicted to be metastable ($E_d < 0.1$ eV/atom) by respective surrogate models (M3GNet for CDVAE and CrystalTextLLM, CHGNet for ours, with CHGNet being more rigorous as evidenced by lower metastability rates). [§]We exclude structures containing $f$-electron in DFT verification while keeping the denominator as all metastable structures.

## 3.4 SELECTION

Last, the selection stage evolves a population of candidate structures that meet the optimization objectives, such as thermodynamic stability or other desired physical properties. For each iteration $i$, we construct a new parent pool $\mathcal{P}_{i+1}$ of the same size ($K \times P$) by selecting top-ranked candidates from three sources: the current parent pool ($\mathcal{P}_i$), newly generated children structures ($\mathcal{C}_i$), and an optional extra pool ($\mathcal{R}$) to improve diversity. Candidates in $\mathcal{P}_i \cup \mathcal{C}_i \cup \mathcal{R}$ are ranked according to optimization objectives. For single-objective optimization, we can select based on either lower decomposition energy $E_d$ (for stability) or higher bulk modulus (as an example for property optimization). For multi-objective optimization, we alternate among multiple objectives, with additional strategies detailed in Appendix E.

## 3.5 FINAL DFT VERIFICATION

After completing all evolutionary iterations, we collect the cumulated offspring structures $\mathcal{S} = \bigcup_i \mathcal{C}_i$ for final validation using Density Functional Theory (DFT). To save computational cost, we focus on meta-stable structures with CHGNet-predicted decomposition energy $E_d < 0.1$ eV/atom. DFT calculations are performed using VASP 6 in the Generalized Gradient Approximation (GGA) with PBE functional [42], using the projector-augmented wave method [32, 33]. We employed a plane-wave basis set with an energy cutoff of 520 eV and a $k$-point mesh of 1,000 per reciprocal atom [28]. The calculations converged to $10^{-6}$ eV in total energy for electronic self-consistent field cycles and 0.02 eV/Å in interatomic forces for the ionic steps. The computational settings are consistent with MPGGARelaxSet and MPGGAStaticSet [27].

## 4 EXPERIMENTS

### 4.1 EXPERIMENTAL SETTINGS

We use Llama 3.1 (70B) [23] as the base LLM. We set temperature to 0.95 to balance creativity and reliability. All experiments use parent size $P = 2$, reproduction size $C = 5$, and $N = 10$ iterations, with population size $K = 100$ unless otherwise specified. Crystal structures are represented in POSCAR format with 12 decimal digits.

**Initialization.** We use the MatBench dataset [19] as the known stable structure set $\mathcal{D}$. From $\mathcal{D}$, we select 3,500 known stable structures as the extra pool $\mathcal{R}$, chosen based on their CHGNet-predicted band gaps closest to 3 eV. This selection criterion biases our pool towards semiconductors and insulators, which often exhibit more diverse and well-defined crystal structures compared to metals. Detailed ablation studies regarding this selection policy are provided in Appendix B.

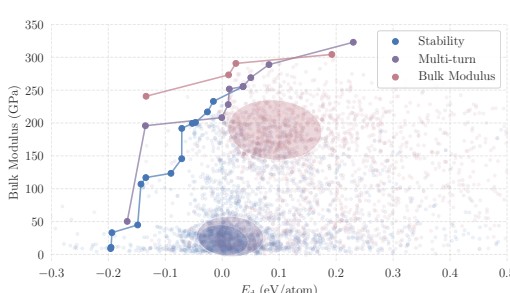
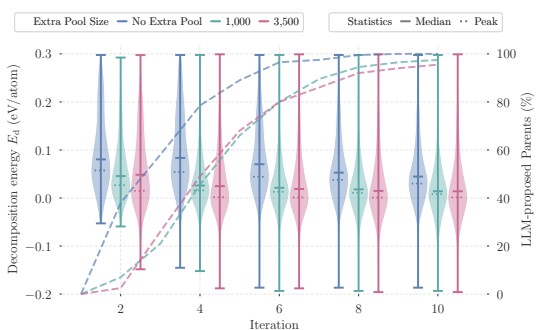
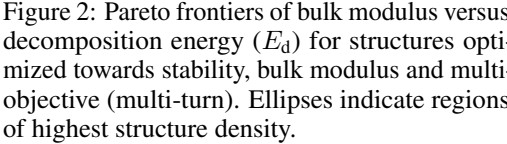

Figure 2: Pareto frontiers of bulk modulus versus decomposition energy ($E_d$) for structures optimized towards stability, bulk modulus and multi-objective (multi-turn). Ellipses indicate regions of highest structure density.

Figure 3: Comparison across different extra pool sizes. (1) Decomposition energy $E_d$ distributions for generated structures (violin plots with solid median and dotted peak lines). (2) Percentage of LLM-proposed structures in the parent pool across iterations (dashed curves).

## 4.2 MAIN EXPERIMENTAL RESULTS

**Crystal structure generation.** We first evaluate the ability of our framework to generate stable crystal structures by optimizing decomposition energy $E_d$ as the sole objective. The LLM prompting template is detailed in Appendix D.

The generation results are reported in Table 1. Following previous work [24, 53], we report structural and compositional validity, which assess non-overlapping atomic radii and charge neutrality respectively. Metastability is evaluated using both CHGNet and M3GNet as surrogate models, measuring the percentage of structures with decomposition energies below 0.1 eV/atom and 0.03 eV/atom thresholds. Structures identified as metastable ($E_d < 0.1$ eV/atom) by CHGNet undergo further DFT calculations for stability assessment.

We compare our model against two baseline models CDVAE [53] and CrystalTextLLM [24]. Among 1,479 generated structures, 76.8% and 81.1% are metastable based on CHGNet and M3GNet evaluations respectively, outperforming the 49.8% metastability rate by M3GNet of the state-of-the-art CrystalTextLLM 70B model, which has a comparable model size to our base model. Under rigorous DFT validation, 31.7% of the metastable structures remain stable, substantially improving the 10.6% stability rate from CrystalTextLLM 70B.

However, structures containing $f$-electron elements (actinides and lanthanides, abbreviated as $f$-ele) lead to challenges in stability prediction due to their strongly correlated electron interactions, which may not be adequately captured by DFT approaches under GGA and Hubbard $U$ corrections [3]. We find that structures with $f$-block elements consistently yield lower decomposition energies ($E_d$), posting a potential computational shortcut in the optimization process. To assess this effect, we report the percentage of stable structures without $f$-ele (denoted as "w/o $f$-ele") among the metastable structures.

Based on this observation, we implemented a mitigation strategy that excludes structures containing $f$-electron elements from being selected as parents. Under this intervention, the metastability rate improves to 78.4%, while the DFT-verified stability slightly decreases to 27.0%. Most notably, the proportion of stable structures without $f$-electrons increases significantly from 14.0% to 24.6%, indicating our approach effectively explores alternative stable configurations. While this computational shortcut remains largely unaddressed by existing methods, our framework demonstrates effective control over structural exploration through simple interventions in the evolutionary process.

While achieving better performance, our method also offers significant computational advantages. Compared to CrystalTextLLM which requires extensive fine-tuning on more than 120K structures, we achieve higher stability rates using only a few reference structures and direct LLM inference. The computational cost is primarily from structure evaluation rather than model training or fine-tuning, making our approach more accessible.

| Extra Pool Size | $E_d < 0.1$ eV/atom | $E_d < 0.03$ eV/atom |
|---|---|---|
| No Extra Pool | 71.25% | 37.45% |
| 1,000 | 80.97% | 58.56% |
| 3,500 | 76.81% | 56.52% |

Figure 4: Metastability rates (percentage of generated structures) under different extra pool size (decomposition energy evaluated by CHGNet at thresholds of $E_d < 0.1$ eV/atom and 0.03 eV/atom).

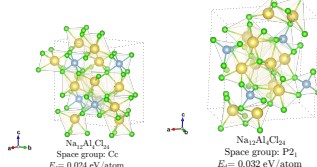

Figure 5: Examples of predicted crystal structures with composition $Na_3AlCl_6$.

**Crystal structure design.** We also explore multi-objective optimization by extending our framework to balance stability with desired material properties. We demonstrate this capability by alternating between optimizing stability ($E_d$) and bulk modulus in each iteration. While this multi-objective setting naturally yields lower stability rates (57.1% metastable with $E_d < 0.1$ eV/atom and 15.6% DFT-verified stable structures with $f$-electron elements) compared to stability-only optimization, it enables the discovery of structures with favorable property-stability trade-offs.

As shown in Figure 2, the Pareto frontiers under various optimization strategies converge in regions with high bulk modulus ($> 200$ GPa) and metastability ($E_d \leq 0.1$ eV/atom) in the stability-property space, indicating successful discovery of potentially valuable structures that balance both objectives. The regions of highest structure density, estimated using Gaussian KDE and visualized as ellipses, reveal how optimization goals affect the distribution. Prioritizing bulk modulus shifts the density distribution toward higher mechanical strength at the cost of increased decomposition energy. We provide additional discussions of property-specific and multi-objective optimization strategies in Appendix E.

**Crystal structure prediction.** We next evaluate our framework on crystal structure prediction tasks, which aim to predict stable structure (i.e. lattice and atomic coordinates) for a given composition. As a case study, we prompt the LLM to predict polymorphs of $Na_3AlCl_6$. For context, the Materials Project database currently contains only one structure for this composition (`mp-1111450`, $Fm\bar{3}m$, $E_d = 0.142$ eV/atom), which is significantly unstable.

During the prompting process, we apply specific structural filters to select seed structures containing only three distinct elements in a 3:1:6 ratio, matching the stoichiometry of $Na_3AlCl_6$. From MatBench, we identified 820 structures meeting these criteria, which formed our initial and extra retrieval pool. Example structures proposed by the LLM for this composition are visualized in Figure 5, with DFT-verified decomposition energies of 0.024 and 0.032 eV/atom respectively. Although these predicted polymorphs remain metastable, their decomposition energies $E_d$ are significantly lower than the previously reported structure in MatBench ($E_d$ reduced by up to 83%), exemplifying the potential of our evolutionary pipeline for CSP applications.

### 4.3 DETAILED ANALYSIS

To better understand the effectiveness of our framework, we conduct a comprehensive analysis by examining three key aspects: the evolution of parent structure quality across iterations, the impact of extra pool size on generation, and the diversity of generated structures. Additional ablation studies on factors affecting generation performance are discussed in Appendices I and J.

**Evolution of parent structure quality.**

Figure 3 illustrates the distribution of decomposition energy and the proportion of LLM-proposed structures using different extra pool sizes. The effectiveness of evolutionary search is demonstrated by the progressive improvement in parent structure quality. We also observe a systematic transition from MatBench-sourced to LLM-generated parent structures across successive generations, regardless of pool size configurations. This growing proportion of LLM-generated structures in the parent pool indicates our framework effectively explores and optimizes the stability landscape.

**Impact of extra pool size.** To evaluate how additional reference structures affect generation performance, we examined three configurations: (1) no extra pool, using only the initial ($K \times P$) randomly

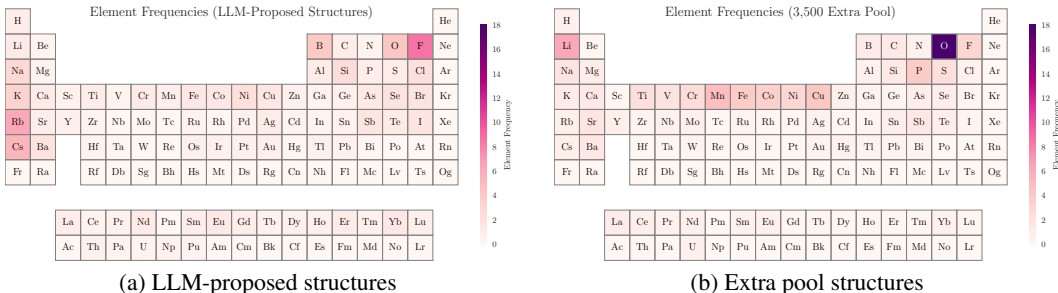

(a) LLM-proposed structures  (b) Extra pool structures

Figure 6: Element frequencies in LLM-proposed structures (left) and extra pool structures (right).

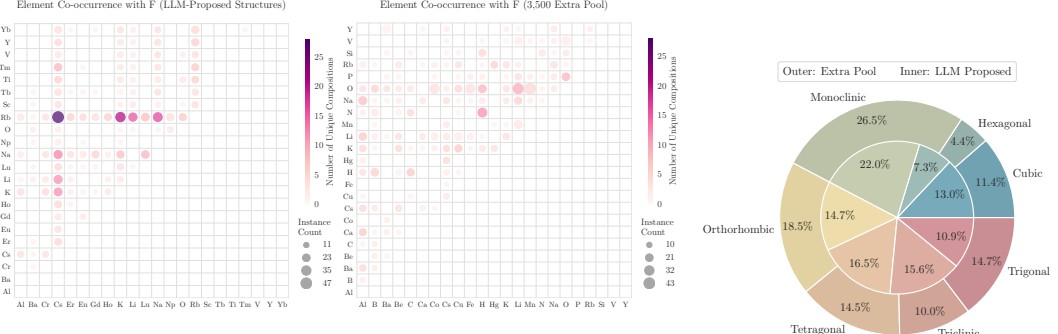

Figure 7: Element co-occurrence patterns with fluorine (F) in LLM-proposed structures (left) versus 3,500 extra pool structures (right). Bubble size indicates frequency of occurrence for each element pair, while color intensity represents compositional diversity (darker indicates more unique compositions with that element pair).

Figure 8: Crystal systems distribution comparison between extra pool of 3,500 structures (outer ring) and LLM-proposed structures (inner pie).

selected structures, (2) an extra 1,000 randomly selected structures, and (3) an extra 3,500 structures retrieved with band gaps closest to 3 eV.

Figures 3 and 4 reveal that introducing a reference pool significantly improves (meta)stability rate, but with diminishing returns for larger pools. The metastability rate ($E_{\mathrm{d}} < 0.1$ eV/atom) increases substantially from 71.25% to 80.97% when adding the 1,000 extra structures, but plateaus with further expansion to 3,500 structures. Beyond stability metrics, each configuration exhibits distinct compositional patterns. Structures generated with no extra pool show diverse combinations with transition metal compounds, while the 1,000 extra pool configuration yields more balanced cation-anion distributions. The 3,500 pool demonstrates a preference for stable fluoride-based compounds, with Cs-F-Rb appearing as the most frequent combination (1.2% occurrence). This shift in compositional preferences suggests that larger pools enable more focused exploration of chemically favorable regions while maintaining structural diversity. Further analyses showing specific crystal structures and detailed compositional diversity across different pool sizes are presented in Figure S3 in Appendix F.

**Structural and compositional diversity.**

To evaluate the diversity of our generated structures, we analyzed their compositional and structural characteristics by comparing LLM-proposed structures and with the extra pool. Figure 6 presents element frequency distributions for both sets. The results show a compositional evolution from predominantly transition metal oxides in reference structures to alkali metals and halogens, with fluorine (F) appearing in 8.6% of the LLM-proposed structures.

Our element co-occurrence analysis reveals high compositional diversity in the LLM-proposed structures, with even the most frequent compositions appearing only twice (approximately 0.14% of total structures). Examination of element co-occurrences with F Figure 7 highlights the effectiveness of our evolutionary method in guiding structure generation toward stable F-based compounds par-

ticularly with alkali metals and transition metals. The structural diversity is further evidenced in Figure 8, which compares crystal system distributions as determined by the SpacegroupAnalyzer from pymatgen [39]. This distribution confirms that our evolutionary method successfully navigates toward stable regions of chemical space while maintaining diverse structural motifs across different crystal systems. Additional diversity and novelty evaluations and analyses are provided in Appendix G.

## 5 RELATED WORK

### 5.1 LANGUAGE MODELS FOR MATERIALS SCIENCE

The increasing capabilities of LLMs have prompted materials science community to explore their potential for understanding and predicting material properties [26]. However, benchmarking studies suggest fine-tuning LLMs over specific materials datasets is necessary to achieve performance comparable to or better than specialized graph neural networks [46]. Research in crystal structure generation has developed along two main paths. Flam-Shepherd & Aspuru-Guzik [21] demonstrate that autoregressive models trained from scratch with character-level tokenization can generate chemically valid crystal structures by directly tokenizing CIF files into string sequences. Secondly, CrystalTextLLM [24] fine-tunes a pre-trained LLM (over massive texts) on generating crystalline structures with task-specific prompts. Mat2Seq [54] converts 3D crystal structures into unique 1D sequences that preserve $SE(3)$ and periodic invariance for language model training. While these approaches produce valid structures, they sacrifice the general conversation capabilities of LLMs due to specialized training or fine-tuning on crystallographic data. In parallel developments within molecular chemistry, MolLEO [51] successfully employs pre-trained LLMs without domain-specific fine-tuning to search for small molecules. Subsequent work [36] extended this evolutionary optimization approach to more complex transition metal chemistry using advanced base LLMs with enhanced reasoning capabilities. However, these applications benefit from natural string representations for molecules (e.g., SMILES or SELFIES), which are considerably simpler than the three-dimensional representations required for crystal structures. Our work bridges this gap by adapting the evolutionary approach to the more complex domain of crystal structures without requiring fine-tuning.

### 5.2 GENERATIVE MODELS FOR MATERIALS DISCOVERY

Besides autoregressive language models, various generative models including variational autoencoders, diffusion models, and flow models have emerged as promising solutions for crystal structure generation. Early work proposes generative crystal structures using variational autoencoders that represent crystal structures as 3D voxels [13, 38]. CDVAE first proposes to generate crystal structures with a score-based generative (diffusion) model and optimize crystal structure properties through gradient-based optimization in the latent space [53]. This approach has been extended in several directions: Jiao et al. [29] developed Riemannian diffusion models to better handle periodic coordinates, Zeni et al. [58] scaled the approach to encompass elements across the entire periodic table with various design criteria, and Dai et al. [15] applied it to crystal inpainting tasks. Most recently, Sriram et al. [48] introduced Riemannian flow matching models to better address periodic boundary conditions with improved performance. Yang et al. [56] explore the synergy between language and generative models by leveraging LLMs to propose chemical formulae under design constraints before feeding them to a diffusion model.

## 6 CONCLUSION

In this paper, we present an evolutionary workflow for computational materials discovery, encompassing crystal structure generation, prediction, and objective-based optimization. We demonstrate that a pre-trained LLM trained on general text can identify a higher proportion of (meta)stable materials compared to state-of-the-art generative models specifically trained on materials datasets. These findings suggest that LLMs inherently function as effective crystal structure generators, with both compositional and structural information naturally embedded within their text inference capabilities. In conclusion, our method complements existing structure discovery techniques by providing refined optimization capabilities while maintaining versatility in addressing various optimization objectives, offering an efficient approach for high-throughput materials discovery.

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

# Supplementary Material for MATLLMSEARCH

## A    ALGORITHMIC WORKFLOW OF MATLLMSEARCH

---
**Algorithm 1** The MATLLMSEARCH Framework

---
**Require:** Population size $K$, parent size $P$, reproduction size $C$, number of iterations $N$, known stable structures $\mathcal{D}$, oracle function $O$, extra pool $\mathcal{R}$

1: ▷ *Initialization*
2: Form population $\mathcal{P}_0$ by sampling $K$ groups of $P$ structures from $\mathcal{D}$
3: Initialize structure collection $\mathcal{S} \leftarrow \varnothing$
4: **for** $i \leftarrow 0, 1, \cdots, (N-1)$ **do**
5:     ▷ *LLM-guided reproduction*
6:     Generate prompts from parent structures in $\mathcal{P}_i$
7:     Obtain offspring structures $\mathcal{C}_i$ via LLM inference and parsing
8:     ▷ *Structure evaluation*
9:     Relax structures $\mathcal{C}_i \leftarrow \text{CHGNetRelax}(\mathcal{C}_i)$
10:     Calculate decomposition energy $E_d$ and properties
11:     Evaluate objective scores using oracle function $O$
12:     Update structure collection $\mathcal{S} \leftarrow \mathcal{S} \cup \mathcal{C}_i$
13:     ▷ *Selection*
14:     Form candidate pool from parents $\mathcal{P}_i$, offspring $\mathcal{C}_i$, and extra pool $\mathcal{R}$
15:     Select top-$(K \times P)$ structures based on objective scores from the candidate pool
16:     Construct next parent groups $\mathcal{P}_{i+1}$
17: Validate final structures via DFT
18: **return** cumulated structures $\mathcal{S}$

---

## B    EXPERIMENTAL DETAILS OF POPULATION INITIALIZATION

The retrieval set $\mathcal{R}$ used consists of 3,500 stable structures sampled from known stable structures–Matbench-bandgap dataset [19], which consists of 106,113 crystal structures in total. To initialize the parent structures for the first iteration, we applied a simple rule-based structure sampling to the structures. First, we checked if the composition was charge-balanced. Second, we verified that for each atom in the crystal structure, there exists at least one valid bond with another site. In addition, we removed structures with simple or overly complicated compositions, i.e., keeping candidate structures with 3 to 6 elements. Finally, we applied random shuffling and de-duplication by composition to the candidate structures. For computational efficiency, we took the top 3,500 structures with a bandgap closest to 3 eV from the pool as extra pool of reference structures during the selection step. The analysis of how the size and sampling rule of the extra pool affect the performance is provided in Section 4.3. To further enhance the structure generation, we envision future work that could explore how structures can be ensembled to form a larger candidate pool for parent selection.

## C    REPRODUCIBILITY

The crystal structures generated by MATLLMSEARCH can be downloaded here. The implementation of our evolutionary search pipeline is available here.

# D    PROMPT FOR CSG

*You are an expert material scientist.  Your task is to propose hypotheses for {reproduction_size} new materials with valid stable structures and compositions. No isolated or overlapped atoms are allowed.*

*The proposed new materials can be a modification or combination of the base materials given below.*

*Format requirements:*

*    1.  Each proposed structure must be formatted in JSON with the following structure:*

```
{{
    "i":  {{
        "formula":  "composition_formula",
        "POSCAR":  "POSCAR_format_string"
    }}
}}
```

*    2. Use proper JSON escaping for newlines (\n) and other special characters*

*Base material structure for reference:*
*    {reference_structures}*

*Your task:*
*    1. Generate {reproduction_size} new structure hypotheses*
*    2. Each structure should be stable and physically reasonable*
*    3. Format each structure exactly as shown in the input*

*Output your hypotheses below:*

# E    ADDITIONAL EXPERIMENTS OF STABLE AND OPTIMIZED CRYSTAL STRUCTURE GENERATION

| Model | $f$-ele in Parents | Validity | | Metastability | | |
| | | Structural | Composition | M3GNet | CHGNet | |
| | | | | $E_d < 0.1$ | $E_d < 0.1$ | $E_d < 0.03$ |
| CDVAE | — | 100.0% | 86.7% | 28.8% | — | — |
| CrystalTextLLM-7B | — | 96.4% | 93.3% | 35.0% | — | — |
| CrystalTextLLM-13B | — | 95.5% | 92.4% | 38.0% | — | — |
| CrystalTextLLM-70B | — | 99.6% | 95.4% | 49.8% | — | — |
| MATLLMSEARCH (Llama 3.1-70B) | Stability | 100.0% | 79.4% | 81.1% | 76.8% | 56.5% |
| | Bulk Modulus | 100.0% | 82.9% | 27.0% | 43.3% | 8.3% |
| | Multi-turn | 100.0% | 84.2% | 70.9% | 57.1% | 29.8% |
| | Weighted sum | 100.0% | 85.1% | 61.8% | 52.3% | 27.4% |

Table S1:  Compare experimental results under various optimization goals.  We explored multi-objective optimization for stability and bulk modulus in two different ways.

The flexibility of our evolutionary pipeline is demonstrated by its ability to guide LLMs in proposing novel crystal structures with diverse mechanical characteristics. We further evaluate model performance under four distinct optimization strategies: (1) stability-oriented optimization ("Stability"),

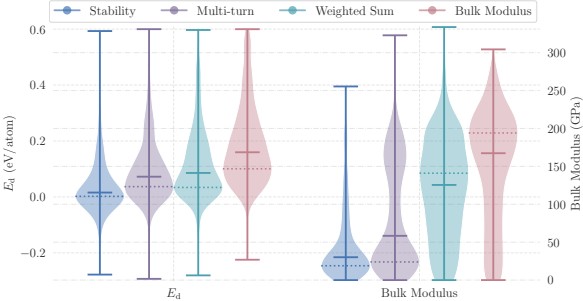

Figure S1: Comparison of optimization strategies targeting different objectives evaluated based on thermodynamic stability (decomposition energy $E_{\rm d}$) and mechanical property (bulk modulus).

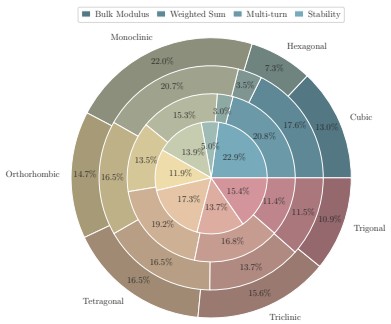

Figure S2: Crystal systems distribution under varied objectives.

(2) property-oriented optimization ("Bulk Modulus"), (3) alternating multi-objective optimization ("Multi-turn"), and (4) weighted-sum optimization ("Weighted Sum"). As shown in Table S1, all four optimization strategies maintain high metastability rates for the proposed structures, which demonstrate that our algorithm can optimize specific properties while maintaining structural validity and stability. Our multi-objective strategies successfully navigate the inherent trade-offs, maintaining reasonable stability while achieving improved mechanical properties.

**Bulk modulus optimization.** To validate the capability of MATLLMSEARCH for property-guided generation, we conduct single-property optimization by modifying the selection criteria from decomposition energy ($E_{\rm d}$) to bulk modulus. In crystalline solids, bulk modulus serves as a key indicator for designing materials with enhanced mechanical hardness. Our experiments used bulk modulus values derived from the Birch-Murnaghan equation of state as a proof of concept. For more comprehensive materials design applications, this approach can be extended to include elastic tensors from DFT calculations or tensorial predictions using equivariant graph neural networks [52].

Figure S1 presents the distribution comparison of decomposition energy ($E_{\rm d}$) and bulk modulus for structures generated under varied optimization strategies, revealing distinct performance trade-offs. The bulk modulus optimization generated more structures with larger bulk modulus values, reaching a peak density at 194 GPa compared to only 19 GPa in stability-oriented optimization. However, this enhancement comes at the cost of increased decomposition energy, with the $E_{\rm d}$ density peaks shifting from 0.0 eV/atom in stability-oriented optimization to 0.1 eV/atom in bulk modulus optimization, indicating reduced thermodynamically stability across iterations.

**Multi-objective optimization.** Beyond single-objective optimization, we explored multi-objective optimization approaches to simultaneously target both thermodynamic stability and mechanical properties using two different multi-objective optimization strategies.

The first approach implements an alternating optimization strategy ("Multi-turn"), where the algorithm alternates between optimizing stability and property in successive iterations. Stability is optimized in the first iteration to set a foundation for property optimization. For customized multi-objective optimization, the number of iterations for each optimization goal can be adjusted. As shown in Figure S1, this method achieves balanced performance in optimizing stability and bulk modulus, with $E_{\rm d}$ centered around 0.037 eV/atom. We observe that bulk modulus distribution separates structures into groups with high mechanical strength at moderate stability versus high stability with lower mechanical strength, suggesting the inherent trade-off in crystal structure generation.

Our second methodology employs a weighted sum approach, combining decomposition energy $E_{\rm d}$ and bulk modulus in a single objective function $\mathcal{J} = 10E_{\rm d} - \text{BulkModulus}$. After sorting the candidate pool by this objective, we select the top structures as parents for subsequent generations. The weighted sum strategy produces crystal structures with bulk modulus centered around 141 GPa and $E_{\rm d}$ densely centered at 0.034 eV/atom. While single-objective stability optimization achieves the highest metastability rate of 76.81%, both multi-objective approaches maintain rates above 50% while enhancing mechanical properties.

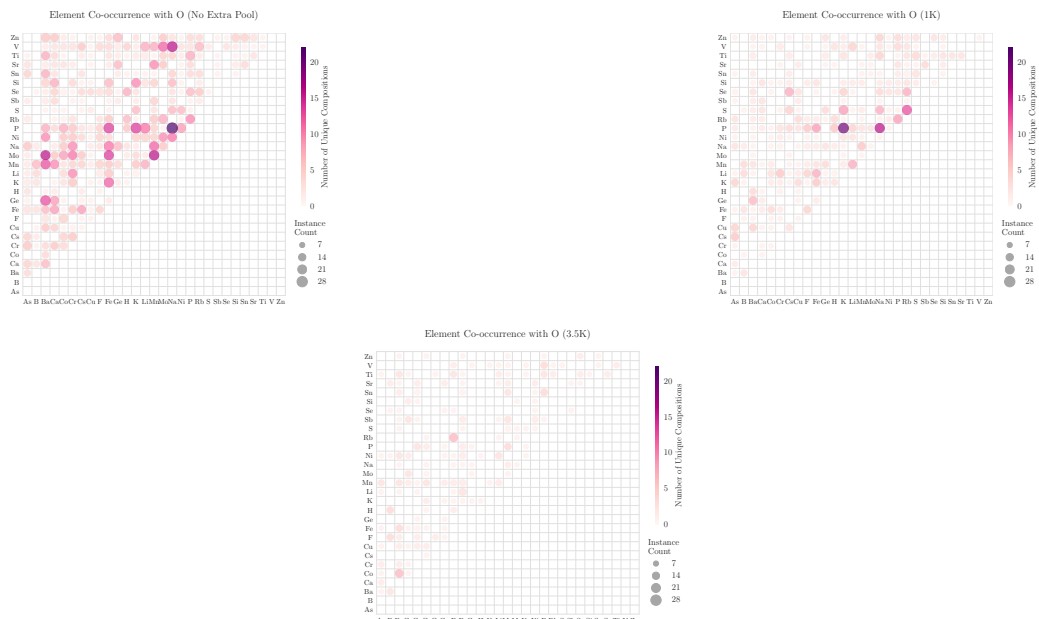

Figure S3: Element co-occurrence patterns with oxygen (O) in LLM-proposed structures across three different extra pool configurations: no extra pool (left), 1,000 random structures (middle), and 3,500 structures with band gaps closest to 3 eV (right). Bubble size represents frequency of occurrence while color intensity indicates compositional diversity.

In addition, the analysis of crystal system distributions in Figure S2 reveals relatively uniform representation across all optimization strategies, indicating that our framework preserves structural diversity regardless of the optimization objective.

## F ANALYSIS OF EXTRA REFERENCE POOLS

In Section 4.3, we examined three configurations of extra pool: (1) no extra pool, using only the initial randomly selected structures ($K \times P$), (2) an extra 1,000 randomly selected structures, and (3) an extra 3,500 structures retrieved with band gaps closest to 3 eV.

**Stability performance.** Our analysis reveals that structure generation achieves optimal metastability rates with a moderate-sized extra pool of reference structures, as demonstrated by the rates of 80.97% and 76.81% with 1,000 and 3,500 extra reference structures, respectively. These results indicate that while additional reference structures improve stability outcomes over using no extra pool (71.25%), the returns diminish as the pool size increases beyond a few thousand structures.

**Evolution of parent source.** As shown in Figure 3, larger extra pools demonstrate more gradual adoption of LLM-generated parents across iterations. This pattern indicates more thorough exploration of the reference space before transitioning to LLM-generated structures, suggesting that larger pools provide a broader foundation for structure generation.

**Compositional diversity.** Analysis of element combinations reveals distinct patterns in LLM-proposed structures across different extra pool configurations. Structures generated with no extra pool show diverse combinations with transition metal compounds, while the 1,000-structure extra pool exhibits more balanced cation-anion distributions. The 3,500-structure pool demonstrates a preference for stable fluoride-based compounds, with Cs-F-Rb appearing as the most frequent combination (1.2% occurrence). Figure S3 illustrates the oxygen-containing compounds proposed by LLMs across the three configurations. With no extra pool or a small extra pool, the LLM tends to propose safer and less novel oxygen-containing compositions. In contrast, larger pools enable greater exploration into chemically diverse spaces, particularly stable fluorine compounds. This shift in

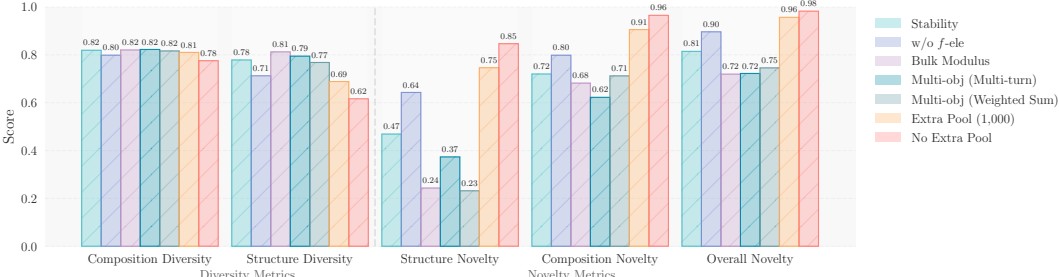

Figure S4: Diversity and novelty evaluation results for structures proposed under different experimental settings.

compositional preferences suggests that larger pools enable more focused exploration of chemically favorable regions while maintaining structural diversity.

## G    EVALUATION ON DIVERSITY AND NOVELTY OF GENERATED STRUCTURES

We quantitatively evaluate the diversity and novelty of structures generated by our framework across configurations using established metrics from prior work [24, 53]. Crystal diversity is measured by computing pairwise distances between their structural and compositional fingerprints. Additionally, we apply log normalization to composition diversity for 0-1 scale standardization. The novelty measures the distance between generated samples and their closest neighbors in the extra pool of reference structures. The structural distance cutoff and composition distance cutoff used for novelty calculation are 0.1 and 2 respectively. To align with previous work, all metrics are computed on structures predicted to be metastable.

The results are summarized in Figure S4. Across different optimization goals, we observe an interesting trade-off between property-specific optimization and novelty, balancing targeted enhancement against chemical space exploration. When optimizing beyond stability alone, such as targeting bulk modulus or performing multi-objective crystal structure design, we observe decreased novelty while diversity remains consistently high across all optimization goals.

Our investigation of extra pool sizes produced a seemingly contradictory finding: smaller reference pools yield higher novelty scores numerically, while larger extra pools lead to structures with distributions better aligned with stable compositions beyond simple oxygen compounds, as analyzed in Appendix F. This apparent contradiction highlights limitations of these metrics in our specific context. Since these metrics primarily measure overlap between training and generated structures, and our extra pools are substantially smaller than typical training datasets used in previous work, they cannot comprehensively characterize the quality of the generated distributions. This underscores the need for more nuanced evaluation metrics that account for the evolutionary nature of our framework and its guided exploration of the chemical space.

## H    IMPACT OF STRUCTURE RELAXATION

To measure the contribution of structural relaxation in our framework, we introduce a quantity $\Delta E$ to represent the energy difference after and before structural relaxation using CHGNet. Figure S5 reveals that the majority of the proposed structures proposed by LLMs exhibit a relatively small $\Delta E$, with 61.1% showing minimal energy changes ($|\Delta E| < 0.5$ eV/atom) during relaxation. This distribution indicates that our framework generates physically meaningful structures that are already close to their local energy minima, requiring only modest refinements through relaxation.

## I    IMPACT OF STRUCTURE STRING FORMATTING

A number of computational methods has emerged for crystal structure generation using machine learning approaches, as shown in Table S2. Most methods represent crystal structures using 3D

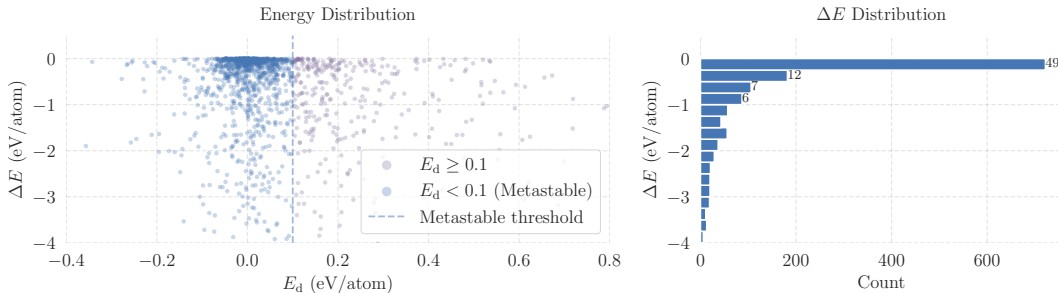

Figure S5: Distribution of energy change $\Delta E$ before/after structural relaxation and decomposition energy ($E_d$) for structures proposed by LLM, evaluated using the pretrained CHGNet.

| Method | Primary Format | Generative | Model | Training |
|---|---|---|---|---|
| CDVAE [53] | 3D | Diffusion | GNN | Training |
| MatterGen [58] | 3D | Diffusion | GNN | Training |
| Flam-Shepherd & Aspuru-Guzik [21] | 3D | AR | Transformer | Training |
| DiffCSP [29] | 3D | Diffusion | GNN | Training |
| CrystalTextLLM [24] | Text/CIF | LLM | Transformer | Fine-tuning |
| FlowMM [48] | 3D | Flow | GNN | Training |
| MATLLMSEARCH (Ours) | Text/CIF/POSCAR | LLM | Llama 3.1 | N/A |

Table S2: A collection of generative models on computational materials discovery. Training denotes if training/fine-tuning is required on materials databases. CSG, CSP, and CSD are abbreviations for three tasks considered (Section 2.1).

information processed through either Graph Neural Networks (GNN) or Transformer architectures, employing various generative strategies like diffusion models or autoregressive approaches. More recently, text-based formats and Large Language Models (LLMs) have emerged as an alternative approach, signaling a promising shift in crystal structure generation and analysis techniques.

The encoding of crystallographic structures into text-based format is essential for LLM processing, making the structural representation an important consideration in our framework design. We investigated the impact of different formatting strategies on generation efficiency and performance: CIF format and POSCAR format with either 4 or 12 decimal places of precision. See Figure S7 for examples.

First, we examine the token efficiency by analyzing the MatBench dataset for token length distribution as shown in Figure S6. The distribution indicates that the POSCAR format with 4 decimal places offers the most token-efficient representation while maintaining reasonable precision, followed by the POSCAR with 12 digits and CIF format. CIF format requires more tokens than POSCAR format, given that CIF uses a more verbose structure and additional metadata.

Performance evaluation shown in Table S3 suggests that POSCAR formatting in 12 decimal places demonstrates slightly better overall performance in the rate of (meta)stability of generated structures under different criteria ($E_d < 0.03$ or $0.1$ eV/atom). Therefore, we employ POSCAR of 12 decimal places as a trade-off results of token efficiency and informativeness. The marginal difference across format may be attributed to the crystallographic data exposed to the LLMs during pre-training. However, it is noteworthy that performance differences across formats remain modest, suggesting the resilience of our approach across different structural representations.

## J HYPER-PARAMETER STUDIES

**Reproduction parameters.** Our training-free evolutionary framework significantly reduces hyper-parameter sensitivity compared to traditional machine learning methods. The reproduction phase introduces several key hyper-parameters that influence LLMs' generation behavior and efficiency,

| Format | # Unique / # Total generated | $E_d < 0.1$ eV/atom | $E_d < 0.03$ eV/atom |
|---|---|---|---|
| POSCAR (4) | 76.7% | 75.4% | 55.3% |
| POSCAR (12) | 72.3% | 76.8% | 56.5% |
| CIF | 75.1% | 68.9% | 49.5% |

Table S3: Proportion of unique structures and their CHGNet-predicted metastability using different structure formats.

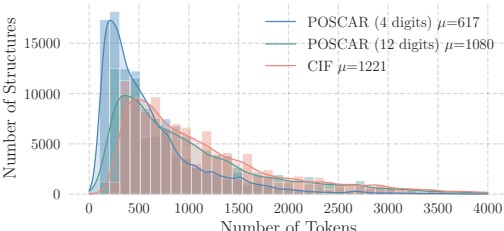

Figure S6: Token efficiency comparison under CIF formatting and POSCAR formatting for the precision of 4 and 12 decimal. $\mu$ indicate the mean of token lengths.

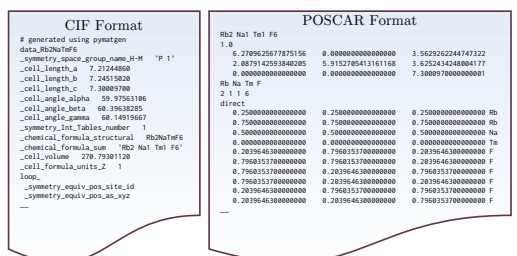

Figure S7: Structure string examples of CIF format and POSCAR format.

including population size ($K$), context size ($C$), and children size ($c$). Our baseline configuration ($C = 2, c = 5$) leverages the Llama 3.1 (70B) model to achieve balanced performance, generating 72.29% unique structures while maintaining high stability rates.

Analysis of parent-to-children ratios reveals that increasing parent diversity ($C = 5, c = 2$) can enhance composition uniqueness of generated structures to 95.49%, though at the price of slight decrease in stability, as presented in Table S4. Conversely, results with single parent demonstrates that crossover between multiple parent structures is beneficial for maintaining structural diversity and stability in the generation process. Overall, we believe that higher parent-to-children ratios can lead to better overall quality in generated structures.

Our analysis also reveals that larger population sizes $K$ can maintain high stability and validity rates comparable to smaller populations. One potential benefit of increasing population size is the diversity introduced in the iteration process, which can alleviate the overpopulation of $f$-ele structures but also lead to higher compositional diversity. However, the increased diversity is offset by higher rates of structural duplication across iterations, suggesting earlier convergence may be needed. Our findings above enable application-specific optimization of the framework's parameters.

**Model temperature.** The temperature hyper-parameter controls sampling randomness in language models by scaling the logits before softmax transformation. Higher temperatures flatten the probability distribution, increasing sampling diversity, while lower temperatures concentrate probability mass on the most likely tokens. While temperature is commonly associated with model creativity, with higher temperatures generally producing slightly more novel outputs [41], this relationship remains an active area of research.

Crystal structure generation is a creative task that requires exploring diverse structural possibilities while maintaining physical validity. We employed an LLM inference temperature of 0.95 in our baseline experiments to facilitate broader structural exploration while maintaining reasonable generation stability. In Table S5, we present the metastability evaluated by CHGNet for structures generated with different LLM temperatures. At the temperature of 0.95, the LLM generated 76.81% metastable structures with $E_d < 0.1$ eV/atom as evaluated by CHGNet. Reducing the temperature to 0.7 maintained robust performance, producing 75.38% metastable structures. Further lowering the temperature to 0.5 yields 71.18% metastable structures. If we choose $E_d < 0.03$ eV/atom as the stability criterion, the percentage of qualifying structures at temperatures 0.95, 0.7, 0.5 and 0.2 are be 56.52%, 56.64%, 51.37% and 50.17% respectively. The consistent high stability rates across temperature settings demonstrate the robustness of our pipeline to LLM hyper-parameter variations.

| Reproduction Configuration | # Unique / # Total generated | $E_d < 0.1$eV/atom | $E_d < 0.03$ eV/atom |
|:---:|:---:|:---:|:---:|
| $1 \rightarrow 5$ | 56.5% | 79.8% | 56.4% |
| $2 \rightarrow 5$ | 72.3% | 76.8% | 56.5% |
| $2 \rightarrow 2$ | 86.3% | 74.8% | 54.3% |
| $5 \rightarrow 5$ | 92.7% | 72.3% | 47.3% |
| $5 \rightarrow 2$ | 95.5% | 68.3% | 46.1% |

Table S4: Proportion of unique structures and their CHGNet-predicted metastability under varying reproduction configurations.

| LLM Temperature | # Unique / # Total generated | $E_d < 0.1$ eV/atom | $E_d < 0.03$ eV/atom |
|:---:|:---:|:---:|:---:|
| 0.95 | 72.3% | 76.8% | 56.5% |
| 0.7 | 70.7% | 75.4% | 56.6% |
| 0.5 | 70.7% | 71.2% | 51.4% |
| 0.2 | 69.8% | 70.3% | 50.2% |

Table S5: Proportion of unique structures and their CHGNet-predicted metastability with different LLM temperatures.

## K  DETAILS OF MACHINE LEARNING INTERATOMIC POTENTIALS

A significant breakthrough in addressing computational cost challenges has emerged through the development of machine learning interatomic potentials (MLIPs) trained based on high-fidelity quantum mechanical calculations (e.g., DFT) [8, 11, 17, 18, 34, 35, 57, 60]. In MLIPs, the total energy is expressed as a sum of atomic contributions, where each atom's energy depends on its local environment including the atomic coordinates and chemical species of neighboring atoms within a cutoff radius:

$$\hat{E} = \sum_i^n \phi(\{\vec{r}_j\}_i, \{C_j\}_i), \quad \hat{\boldsymbol{f}}_i = -\frac{\partial \hat{E}}{\partial \boldsymbol{r}_i}, \quad \boldsymbol{\sigma} = \frac{1}{V}\frac{\partial \hat{E}}{\partial \boldsymbol{\varepsilon}}. \quad \text{(S1)}$$

Here, $\phi$ is a learnable function that maps the set of position vectors $\{\vec{r}_j\}_i$ and chemical species $\{C_j\}_i$ of the neighboring atoms $j$ to the energy contribution of atom $i$. The forces $\boldsymbol{f}_i$ and stress $\boldsymbol{\sigma}$ are calculated via auto-differentiation of the total energy with respect to the atomic Cartesian coordinates and strain. Recent advances have demonstrated that MLIPs, trained on extensive density functional theory (DFT) calculations accumulated over the past decade across diverse materials systems, exhibit remarkable transferability in performing atomistic simulations across various material and chemical systems. These broadly applicable potentials are known as universal MLIPs (uMLIPs) [7, 10, 16, 40]. By leveraging uMLIPs as surrogate energy models, researchers can rapidly optimize crystal structures and obtain structure-energy relationships for assessing thermodynamic stability. By leveraging uMLIPs as surrogate energy models, one can rapidly optimize crystal structure and obtain the structure-energy relationships for assessing thermodynamic stability. Recent benchmark studies, including MACE [7], DPA-1 [59] and JMP (joint multi-domain pretraining) [47], have demonstrated the high accuracy of these uMLIPs in predicting crystal thermodynamical stability, particularly for industrial-scale implementations trained on millions of compounds and non-equilibrium atomic configurations [6, 37, 55].

To accelerate the oracle function evaluation in the evolutionary iterations, we performed all structure relaxations with the FIRE optimizer [9] over the potential energy surface provided by CHGNet, where the atom positions, cell shape, and cell volume were optimized to reach converged interatomic forces of 0.1 eV/atom [16]. The output energy prediction is directly compatible with the Materials Project phase diagrams with the MaterialsProject2020Compatibility [50].

## L  LIMITATIONS AND FUTURE WORK.

Our study serves as a proof-of-concept and requires further validation in real-world materials discovery workflows. While we have demonstrated that LLM inference is a powerful tool for searching

materials under thermodynamic stability guidance, the practical realization of new materials remains challenging, particularly in terms of successful synthesis.

One limitation observed in our CSP tasks is that the generated structures exhibit similarities to the provided reference structures. The evolutionary nature of the genetic algorithm naturally favors incremental modifications over radical structural changes. Additionally, LLMs exhibit an inductive bias toward known stable structures, often resorting to their pre-trained knowledge and simple atomic substitutions. Nevertheless, our approach can serve as an effective optimization tool in addition to the suggestion of novel structural prototypes, which can be more readily obtained through alternative methods, including variational autoencoders [45, 61], diffusion models [58], random structure searching [43], or response-matching approaches [12]. However, the capability of these methods for comprehensive materials discovery across diverse chemical spaces remains under-explored. In addition, it is an open question that whether the LLM-proposed materials design hypotheses are free of intellectual property issues.

Looking forward, a natural extension of this work would be synthesis prediction based on the evolutionary method. Improved machine learning interatomic potentials will complement this process, as discussed in Appendix K. Such development would benefit from integration with high-quality experimental data from automated, high-throughput experiments, bridging the gap between computational predictions and experimental synthesis, which would accelerate high-throughput materials discovery.

