# OpenReview forum: "Large Language Models Are Innate Crystal Structure Generators"
_ICLR.cc/2025/Workshop/AgenticAI — ICLR 2025 Workshop AgenticAI Oral_

### Official Review · Reviewer_ScPH · 2025-03-02
**LARGE LANGUAGE MODELS ARE INNATE CRYSTAL STRUCTURE GENERATOR**

**Rating:** 7
**Confidence:** 5

**Review:**

1. Demonstrates LLMs as effective crystal structure generators without fine-tuning, reducing computational overhead.

2. Introduces MATLLMSEARCH, an LLM-driven evolutionary search framework for materials discovery.

3. Outperforms specialized models (e.g., CrystalTextLLM), achieving higher metastable and DFT-verified stable structures.

4. Applies evolutionary search to guide LLMs in generating structurally valid and thermodynamically stable crystals.

5. Expands beyond generation to prediction and multi-objective optimization (e.g., bulk modulus, deformation energy).

6. Validates results using machine learning interatomic potentials (MLIPs) and DFT calculations.

7. Employs a structured workflow—Selection, Reproduction, and Evaluation—to iteratively refine structures.

8. Benchmarks against diffusion-based and flow-based generative models but lacks reinforcement learning-based comparisons.

9. Computational efficiency not fully explored—impact of LLM inference costs vs. traditional fine-tuned approaches unclear.

10. Structure evaluation relies on CHGNet, with limited generalization tested across different MLIP methods.

11. No ablation on LLM scale dependency—performance of smaller LLMs vs. 70B models is underexplored.

12. Unclear adaptability to experimental validation and real-world synthesis feasibility.

13. Potential biases in LLM-generated structures—may favor known compositions over truly novel discoveries.

14 Does not explore hybrid approaches integrating fine-tuned models with generative LLMs.

15. Promising results but needs more efficiency analysis, experimental validation, and expanded comparisons.

---

### Official Review · Reviewer_H1nZ · 2025-03-02
**interesting paper**

**Rating:** 7
**Confidence:** 2

**Review:**

This paper proposed a framework MATLLMSEARCH which integrates prerained LLMs with evolutionary search algorithms.
The paper included extensive empirical studies on different tasks to illustrate the insights.

---

### Official Review · Reviewer_hm3J · 2025-03-04
**Large Language Models Are Innate Crystal Structure Generators**

**Rating:** 6
**Confidence:** 2

**Review:**

The paper Large Language Models Are Innate Crystal Structure Generators presents MATLLMSEARCH, an evolutionary framework that combines pre-trained LLMs with evolutionary search algorithms to generate and optimize crystal structures without requiring fine-tuning. Unlike conventional generative models that demand extensive training on materials-specific datasets, MATLLMSEARCH exploits LLM inference alone, demonstrating that pre-trained LLMs inherently encode sufficient chemical and structural knowledge for stable crystal generation. By iteratively selecting, reproducing, and evaluating structures, the framework integrates LLM-generated candidates with thermodynamic validation using CHGNet and DFT calculations, achieving a 76.81% metastable generation rate and 31.7% DFT-verified stability, surpassing fine-tuned models like CrystalTextLLM while significantly reducing computational costs. Beyond structure generation, MATLLMSEARCH extends to crystal structure prediction and multi-objective optimization, optimizing properties like bulk modulus and facilitating the discovery of novel metastable polymorphs. While the paper is well-written, the evaluation primarily compares MATLLMSEARCH against a single baseline (CrystalTextLLM), and additional comparisons would strengthen the analysis. Additionally, clarification on the meaning of "-" in Table 2 would be helpful. Despite eliminating fine-tuning overhead, the framework's effectiveness remains highly dependent on prompt engineering, and the practical synthesizability of the generated structures is yet to be verified, highlighting the need for experimental validation to bridge computational predictions with real-world materials discovery.

---

### Decision · Program_Chairs · 2025-03-05

Accept (Oral)